# DifNet: Semantic Segmentation by Diffusion Networks

**Peng Jiang** [1]        **Fanglin Gu** [1]        **Yunhai Wang** [1]

**Changhe Tu** [1]        **Baoquan Chen** [2,1]

[1]Shandong University, China        [2]Peking University, China

sdujump@gmail.com, fanglin.gu@gmail.com, cloudseawang@gmail.com
chtu@sdu.edu.cn, baoquan.chen@gmail.com

## Abstract

Deep Neural Networks (DNNs) have recently shown state of the art performance on semantic segmentation tasks, however, they still suffer from problems of poor boundary localization and spatial fragmented predictions. The difficulties lie in the requirement of making dense predictions from a long path model all at once, since details are hard to keep when data goes through deeper layers. Instead, in this work, we decompose this difficult task into two relative simple sub-tasks: seed detection which is required to predict initial predictions without the need of wholeness and preciseness, and similarity estimation which measures the possibility of any two nodes belong to the same class without the need of knowing which class they are. We use one branch network for one sub-task each, and apply a cascade of random walks base on hierarchical semantics to approximate a complex diffusion process which propagates seed information to the whole image according to the estimated similarities.

The proposed DifNet consistently produces improvements over the baseline models with the same depth and with the equivalent number of parameters, and also achieves promising performance on Pascal VOC and Pascal Context dataset. Our DifNet is trained end-to-end without complex loss functions.

## 1  Introduction

Semantic Segmentation who aims to give dense label predictions for pixels in an image is one of the fundamental topics in computer vision. Recently, Fully convolutional networks (FCNs) proposed in [1] have proved to be much more powerful than schemes which rely on hand-crafted features. Following FCNs, subsequent works [2–16] have get promoted by further introducing atrous convolution, shortcut between layers and CRFs post-processing.

Even with these refinements, current FCNs based semantic segmentation methods still suffer from the problems of poor boundary localization and spatial fragmented predictions, because of following challenges: First, to abstract invariant high level feature representations, deeper models are preferred, however, the invariance character of features and increasing depth of layers may lead detailed spatial information lost. Second, given this long path model, the requirement of making dense predictions all at once makes these problems more severe. Third, the lack of ability to capturing long-range dependencies causes model hard to generate accuracy and uniform predictions [17].

To address these challenges, we relieve the burden of semantic segmentation model by decomposing semantic segmentation task into two relative simple sub-tasks, seed detection and similarity estimation, then diffuse seed information to the whole image according to the estimated similarities. For each sub-task, we train one branch network respectively and simultaneously, therefore our model has two branches: seed branch and similarity branch. The simplicity and motivation lie in these

following aspects: For seed detection, we hope it can give initial predictions without the need of wholeness and preciseness, this requirement is highly appropriate to the property of DNNs' high level features which are good at representing high level semantic but hard to keep details. For similarity detection, we intend to estimate the possibility of any two nodes that belong to the same class, under this circumstance, relatively low level features already could be competent.

Based on the motivations mentioned above, we let seed branch predict initial predictions and let similarity branch estimate similarities. To be specific, seed branch firstly generates a score map which assigns score value for each class at each node (pixel), then an importance map is learned to re-weight score map to get initial predictions. At the same time, similarity branch will extract features from different semantic levels, and compute a sequence of transition matrices correspondingly. Transition matrices measure the possibility of random walk between any two nodes, with our implementation they could also reflect similarities on different semantic levels. Finally, we apply a cascade of random walks based on these transition matrices to approximate a complex diffusion process, in order to propagate seed information to the whole image according to the hierarchical similarities. In this way, the inversion operation of dense matrix in the diffusion process could be avoided. Our diffusion process by cascaded random walks shares the similar idea as residual learning framework [18] who eases the approximation of a complex objective by learning residuals. Moreover, our random walk actually computes the final response at a position as a weighted sum of all the seed values which is a non-local operation that can capture long-range dependencies regardless of positional distance. Besides, from Fig. 1(a), we can see the cascaded random walks also increase the flexibility and diversity of information propagation paths.

Our proposed DifNet is trained end-to-end with common loss function and no post-processing. In experiments, our model consistently shows superior performance over the baseline models with the same depth and with the equivalent number of parameters, and also achieves promising performance on Pascal VOC 2012 and Pascal Context datasets. In summary, our contributions are:

- We decompose the semantic segmentation task into two simple sub-tasks.
- We approximate a complex diffusion process by cascaded random walks.
- We provide comprehensive mechanism studies.
- Our model can capture long-range dependencies.
- Our model demonstrates consistent improvements over various baseline models.

## 2    Related Work

Many works [2, 8, 9, 15, 14, 13, 12, 11, 10, 19–21] (Here, we mainly focus on methods that are based on deep neural networks, as these represent the state-of-the-art and are the most relevant to our scheme.) have approached the problems of poor boundary localization and spatially fragmented predictions for semantic segmentation.

Currently, conditional random field (CRF) is one of the major approaches used to tackle these two problems. Works, such as [2], use CRF as a disjoint post-processing module on top of the main model. Because of disjoint training and post-processing, they often fail to capture semantic relationships between objects accurately and thus produce segmentation results spatially disjoint. Instead, works [8, 10, 12, 9, 20] propose to integrate CRF into the networks, so as to enable end-to-end training of the joint model. However, this integration may lead to a dramatic increase of parameters and computing complexity that the model usually needs many iterations of mean-field inference or a recurrent scheme [12] to optimize the CNN-CRF models. To avoid iterative optimization, works [13, 14] employ Gaussian Conditional Random Fields which can be optimized by only solving a system of linear equations, but at the cost of increasing complexity of gradient computation. Apart from the view of CRF, work [15] utilizes graphical structures to refine results by random walks, but the calculation has a dense matrix inversion term which is not appropriate for the networks. Unlike above mentioned methods, work [21] does not compute global pairwise relations directly. It predicts four local pairwise relations along the different direction to approximate the global pairwise relations, however, it also leads to the complexity of the model.

To integrate CRF into the model, several works also employ networks with two branches that one for pairwise term and one for unary term. However, the definition and computation are different from

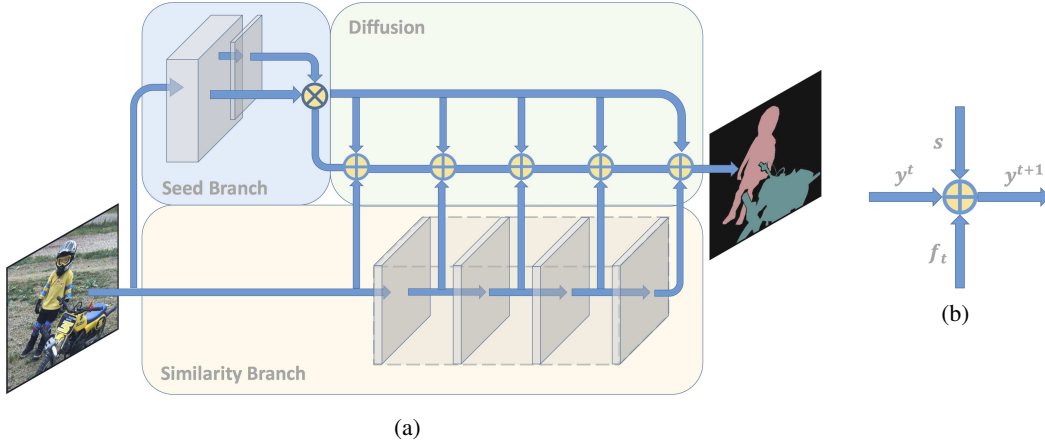

(a)

(b)

Figure 1: (a) Our DifNet contains two branches: 1. Seed Branch, which produces score map and importance map, from which seed are obtained by Hadamard product $\otimes$; 2. Similarity Branch, which extracts features from different layers to compute transition matrices and estimate pixel-wise similarities. Finally, the model approximates the diffusion process by a cascade of random walks $\oplus$ to propagate seed information to the whole image according to the estimated similarities. (b) Random Walk operation. For each random walk, the inputs are: 1. Output of last random walk operation; 2. Features from Similarity Branch; 3. Seed from Seed Branch. Given the inputs, output is calculated by $\oplus$. The specific computation procedure will be explained in Sec. 4.

ours that we represent pairwise similarities by the transition matrix whose summation in each row or column equals one. For the purpose of measuring similarity, different metrics are presented, ours and [14] compute similarities by inner product while most of the others use Mahalanobis distance. It is important to note that our DifNet consists several transition matrices which are computed based on features from different semantic levels, and each random walk operation is conducted according to one transition matrix, see Fig. 1(a). In this way, we do not require each transition matrix contains all the similarity information, which relieves the burden of similarity branch. Besides, the cascaded random walks will also increase the flexibility and diversity of information propagation paths.

For supervised semantic segmentation tasks, cross-entropy is the most common used loss function. However, in some previously mentioned models [15, 20], another loss for similarity estimation had been applied, where similarity groundtruth are transformed from labels. According to [22], optimal pairwise term and optimal unary term are mutually influenced, so strict constraint on only one of these two terms may not lead to good results. Consequently, in our DifNet, we only penalize final predictions by cross-entropy loss. As for training strategy, we train our two branches simultaneously, while some works, for instance [14], train the two branches alternatively.

## 3   Methodology

Given the input image $I$ of size $[c, h, w]$, the pairwise relationship can be expressed as affinity matrix $W_{N \times N}$, where $N = h \times w$ and each element $W_{ij}$ encodes the similarity between node $i$ and node $j$. As mentioned in Sec. 1, our seed vector is defined as $s = Mx$ where $x$ is the score map of size $[N, K]$ ($K$ is the number of classes) and importance map $M$ is the diagonal matrix with the size of $N \times N$ and value in $[0, 1]$.

Assuming the final predictions are $y$, in order to diffuse seed value to all the other nodes according to the affinity matrix, we can optimize the following equation:

$$y = \underset{y^*}{\operatorname{argmin}} \frac{1}{2}(\mu \sum_{i,j=1}^{N} W_{ij} \| \frac{y_i^*}{\sqrt{d_{ii}}} - \frac{y_j^*}{\sqrt{d_{jj}}} \|_2 + (1 - \mu) \sum_{i=1}^{N} M_{ii} \| y_i^* - x_i \|_2) \tag{1}$$

Eq. 1 is convex and has a closed-form solution, without loss of generality:

$$y = (D^{-1}(D - \mu W))^{-1}(1 - \mu)Mx \tag{2}$$

where $D$ is the degree matrix that defined as $D = \text{diag}\{d_{11}, ..., d_{NN}\}$ and $d_{ii} = \sum_j W_{ij}$, $\mu$ is the weight to balance smooth term and data term. Eq. 2 is usually considered as diffusion process that $y = L^{-1}s$, where $s = Mx$ is the seed vector and $L^{-1} = (D^{-1}(D - \mu W))^{-1}$ is the diffusion matrix ($L$ equals to the inversion of normalized graph Laplacians). Works such as [15, 14] propose to use networks with two branches to predict these two parts respectively. However, to compute final predictions $y$, they have to solve dense matrix inversion or a system of linear equations, which is time-consuming and unstable (the matrix to be inverted may be singular.). To tackle this problem, we propose to use a cascade of random walks to approximate the diffusion process. A random walk with the seed vector as initial state is defined as:

$$y^{t+1} = \mu P y^t + (1 - \mu)s \tag{3}$$

where $\mu$ is a parameter in $[0, 1]$ that controls the degree of random walk to other states from the initial state, and $P = D^{-1}W$ is transition matrix whose element measures the possibility that a random walk occurs between corresponding positions and has value in $[0, 1]$. It is important to note that Eq. 3 does not contain dense matrix inversion anymore and is equal to Eq. 2 when $t \to \infty$, which is proved by the following.

*Proof.* Eq. 3 can be reformulated as $y^{t+1} = (\mu P)^{t+1}s + (1 - \mu)\sum_{i=0}^{t}(\mu P)^i s$ by unrolling the recurrence. When $t \to \infty$ and in view of $P, \mu \in [0, 1]$, apparently $\lim_{t \to \infty}(\mu P)^{t+1} = 0$ and $y^{t+1} = (1 - \mu)\sum_{i=0}^{t}(\mu P)^i s$. By computing $y^{t+1} - \mu P y^{t+1}$ and setting $t \to \infty$, we finally obtain $y^{\infty} = (1 - \mu P)^{-1}(1 - \mu)s$ which equals to Eq. 2. $\square$

## 4 Implementation

In this section, we describe how we implement a cascade of random walks by DifNet to approximate the diffusion process.

The key part of the random walk is the computing of transition matrix $P = D^{-1}W$ which is equivalent to conducting a $\text{softmax}$ along each row of $W$ that $P = \text{softmax}(W)$. To compute $W$ and measure similarities between nodes, we apply the inner product of features $f$ from similarity branch. Before that, in our implementation, we first adjust feature dimensions to fixed length by one $conv(1 \times 1)\text{-}bn\text{-}pooling$ layer $\Psi$, so that $W = \Psi(f)^T\Psi(f)$. Because $P$ has encoded similarities between any node pairs and random walk is a non-local operation, so that long-range dependencies can be captured.

To let DifNet approximate the diffusion process by learning, we should take full advantage of the learning capacity of model. Instead of using a predefined and fixed parameter $\mu$, our model learns this parameter and determines the degree of each random walk adaptively. Besides, for the different random walk, we compute the corresponding transition matrix $P_t$ based on features from the different layer of similarity branch which represent different semantic levels, thus the information will be propagated gradually according to different levels' semantic similarities. By this manner, the diffusion process is not merely approximated by the cascaded random walks, but by the cascaded random walks on hierarchical semantics. We demonstrate how these transition matrices look like in Sec. 5.3. Consequently, our random walks can be defined as:

$$y^{t+1} = \mu_t P_t y^t + (1 - \mu_t)s \tag{4}$$

As mentioned before, our seed $s$ is expressed as a multiplication of importance map $M$ and score map $x$, this operation is denoted as $\otimes$ in Fig. 1(a). Score map $x$ is the direct output of seed branch with value in $\mathbb{R}$, thus our DifNet is actually diffusing score and the influence of node $i$ to others in channel $k$ when diffusion should be defined as $|x_{i,k}|$. To further adjust the influence of nodes, we introduce several layers $H$ (with $\text{sigmoid}$ as the last activation) on top of $x$ to predict an importance map $M$ (diagonal matrix), such that $M = H(x)$. Finally, we apply the important map to score map and obtain seed by $s = Mx$. From experiments, we observe that importance map adjusts the influence of nodes base on scores of the neighborhood. Fig. 2 demonstrates score map $x$, influence map $E$ and importance map $M$. Clearly, $M$ will reduce the influence of over-emphasis nodes and outliers. Please see Sec. 5.3 for details.

It's worth to note that in our implementation $s$ is of size $[h' \times w', K]$ and $P$ is of size $[h' \times w', h' \times w']$, where $h' = h/5$ and $w' = w/5$, meanwhile random walks only involve matrix multiplication

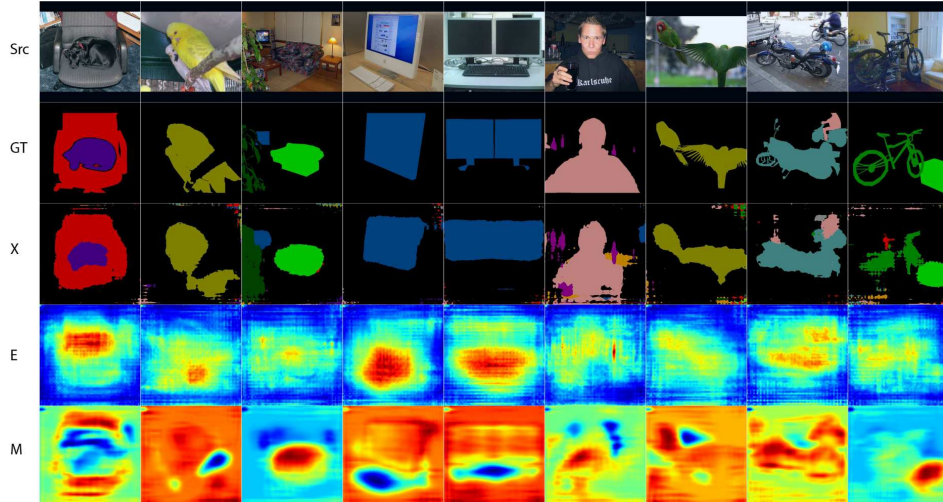

Figure 2: Visualization of input image, groundtruth, score map $x$ (only show class corresponding to the largest score value in each position), influence map $E(x)$ and importance map $M(x)$.

operations, consequently the amount of computation related to random walks in our model will be relatively less.

According to [22], optimal seed $s$ and optimal diffusion matrix $L^{-1}$ are mutually determined. Therefore, we choose to let model learn seed and similarities on its own instead of providing supervision on affinity as [15]. However, no supervision for the cascaded random walks may also cause a problem. By the definition of Eq. 4, if certain $P_t$ cannot gain useful similarity information from corresponding features, $\mu_t$ will be set to a small value by model during training, thereby $y^{t+1} \approx s$. In this case, all the previous results of random walks will be discarded. To preserve useful information of preceding random walks, we propose to employ an adaptive identity mapping term further. By reformulating Eq. 4 as $y^{t+1} = R(y^t, P_t, s, \mu_t)$, finally the $\oplus$ operation can be defined as:

$$y^{t+1} = \beta_t R(y^t, P_t, s, \mu_t) + (1 - \beta_t)y^t \tag{5}$$

where $\beta_t$ is another parameter to be learned, for the sake of controlling the degree of identity mapping. In our experiments, DifNet occasionally assigns a minimal value to certain $\mu_t$, but will also reduce the amount of $\beta_t$ at the same time. In this way, the effect actually equals to omit certain random walk $\oplus$, so the information from preceding random walks can be preserved and passed to following random walks.

Fig. 1(a) shows the whole framework of our DifNet, the upper branch is seed branch while the lower branch is similarity branch. We use $\oplus$ to represent the random walk operation, as illustrated in Fig. 1(b), for each $\oplus$ the inputs are: (1) features $f_t$ from certain intermediate layer of similarity branch ; (2) seed vector $s$ from seed branch; (3) output $y^t$ of previous random walk. Given the inputs, $\oplus$ computes $P_t = \text{softmax}(\Psi(f_t)^T \Psi(f_t))$, determines $\mu_t$ and $\beta_t$, and finally gives output $y^{t+1}$ according to Eq. 5.

## 5 Experiments

### 5.1 Experimental Settings

Our DifNet can be built on any FCNs-like models. In this paper, we choose DeeplabV2 [3] as our backbone. Original DeeplabV2 has reported promising performance by introducing atrous convolution, ASPP module, multi-scale inputs with max fusion, CRF-postproposing, and MS-COCO pretrain. Among these operations, the last three are external components that other models could also benefit from them. Thus, to better study diffusion property of our DifNet, we design our backbone only with atrous convolution, and ASPP module for seed branch.

|                      | mIOU(Val) | mIOU(Test) |
|----------------------|-----------|------------|
| Sim-Deeplab-18       | 66.33%    | -          |
| Sim-Deeplab-34       | 69.76%    | -          |
| Sim-Deeplab-50       | 70.78%    | -          |
| Sim-Deeplab-101      | 71.83%    | 72.54%     |
| Sim-Deeplab-101-CRF  | 72.26%    | -          |
| DifNet-18            | 70.17%    | 70.46%     |
| DifNet-34            | 71.84%    | 71.62%     |
| DifNet-50-noASPP     | 72.52%    | -          |
| DifNet-50            | 72.57%    | 72.55%     |
| DifNet-101           | 73.22%    | 73.21%     |

Table 1: Comparison with simplified DeeplabV2 of different depth on Pascal VOC dataset.

DeeplabV2 are based on ResNet [18] architecture. To approximate the diffusion process and for the sake of efficiency, in view of backbone architecture, in our DifNet we conduct five random walks in total based on five transition matrices computed from features out of four ResNet blocks as well as input.

We study the performance and mechanism of our DifNet on the prevalent used Augmented Pascal VOC 2012 dataset [23, 24] and Pascal Context dataset [25]. Augmented Pascal VOC 2012 dataset has 10,582 training, 1,449 validation, and 1,456 testing images with pixel-level labels in 20 foreground object classes and one background class, while Pascal Context has 4998 training and 5105 validation images with pixel-level labels in 59 classes and one background category. The performance is measured in terms of pixel intersection-over-union (IOU) averaged across all the classes. To train our model and baseline models, we use a mini-batch of 16 images for 200 epochs and set learning rate, learning policy, momentum and weight decay same as [3]. We also augment training dataset by flipping, scaling and finally cropping to $321 \times 321$ due to computing resource limitation.

## 5.2 Performance Study

**Pascal VOC** For quantitative comparison, we use simplified DeeplabV2 models as our baselines which only have atrous convolution and ASPP module as our model. Besides, both our DifNet and baseline models use pre-trained ResNet architecture on ImageNet [26], while other components in the models are trained from scratch. Though our DifNet has two branches, the depth of the model is equal to the deepest one, because data flows through two branches parallelly other than cascadely when doing inference. To be more fair, in Table. 1, instead of making comparison based on the same depth, we also report results based on the equivalent number of parameters. For example, DifNet-50 has the same depth as Sim-Deeplab-50 while has the equivalent number of parameters as Sim-Deeplab-101. In experiments, our models achieve consistent improvements over sim-Deeplab models of the same depth and number of parameters on Pascal VOC validation dataset, and the performance is also verified on the testing dataset. To verify the effectiveness of our diffuse module, we further conduct another two experiments: Firstly, we test DifNet-50 without ASPP module (DifNet-50-noASPP), from experiments we can see ASPP module only plays a limited role. Then, we run Sim-Deeplab-101 with CRF post-processing, which improves the performance from 71.83% to 72.26% at the cost of about 1.8s/image (10 iterations), but is still worse than our DifNet-50 (72.57%).

**Pascal Context** In Table. 2, we further make comparison among DifNet with different depth and components, original DeeplabV2 [3] with different components and other methods on Pascal Context dataset. Compared with baseline models, our DifNet model achieves promising performance by fewer components.

## 5.3 Mechanism Study

In this section, we focus on the mechanism and effect of components in our model. We use DifNet-50 trained on Pascal VOC dataset to carry out following experiments.

| | MSC | COCO | ASPP | CRF | Diffuse | mIOU(Val) |
|---|---|---|---|---|---|---|
| FCN-8s[1] | | | | | | 39.1% |
| CRF-RNN[12] | | | | | | 39.3% |
| ParseNet[6] | | | | | | 40.4% |
| ConvPP-8s[16] | | | | | | 41.0% |
| UoA-Context+CRF[8] | | | | | | 43.3% |
| **ResNet-101** | | | | | | |
| Deeplab[3] | ✓ | | | | | 41.4% |
| Deeplab[3] | ✓ | ✓ | | | | 42.9% |
| Deeplab[3](Sim-Deeplab) | | | ✓ | | | 43.6% |
| Deeplab[3] | ✓ | ✓ | ✓ | | | 44.7% |
| Deeplab[3] | ✓ | ✓ | ✓ | ✓ | | 45.7% |
| DifNet(our model) | | | ✓ | | ✓ | 46.0% |
| **ResNet-50** | | | | | | |
| DifNet(our model) | | | | | ✓ | 44.7% |
| DifNet(our model) | | | ✓ | | ✓ | 45.1% |

Table 2: Comparison with other methods and DeeplabV2 with different components on Pascal Context dataset.

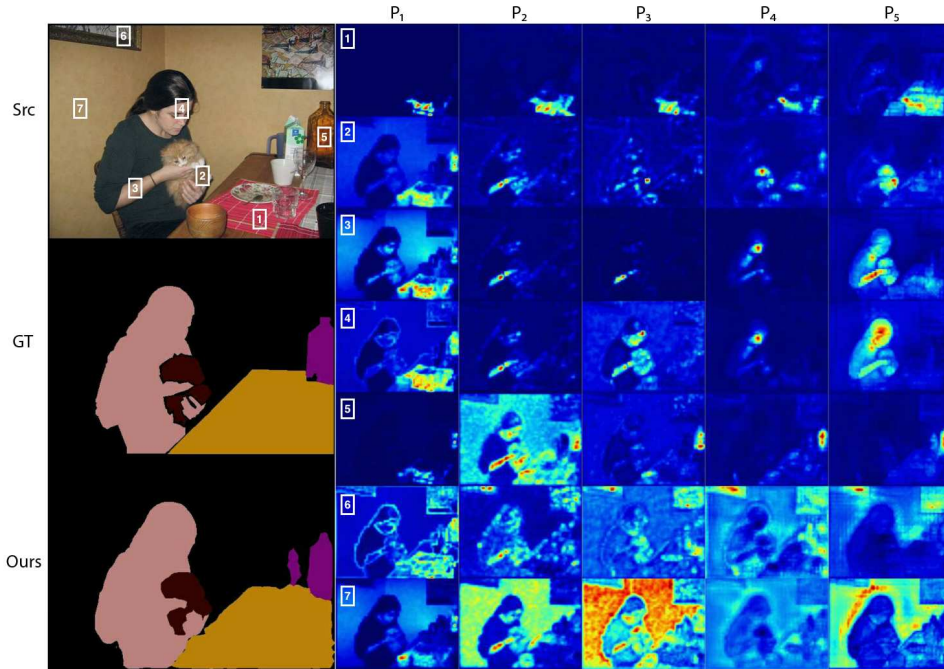

Figure 3: Visualization of corresponding rows in $P_t$ of selected nodes in the image. The right five columns demonstrate similarities measured on different $P_t$, respectively. Nodes more highlighted are more similar to the selected node.

**Seed Branch** We compute seed as $s = Mx$, where $x$ is the score map and $M$ is the importance map learned based on the neighborhood of $x$. The influence of node $i$ in the diffusion process in channel $k$ is $|x_{i,k}|$. To visualize the influence of nodes on all channels, we define influence map $E$ as $E_i = \sum_{k=1}^{K} |x_{i,k}|$. We show our $x$, $E$ and $M$ in Fig. 2. Obviously, $x$ contains many outliers and has the problems of poor boundary localization and spatial fragmented predictions. However, observed from the influence map $E$, most of these outliers have little influence to the diffusion process. The importance map $M$ will further reduce or increase the influence of certain regions, such as columns $4, 5$ where keyboard is suppressed and columns $3, 9$ where sofa is enhanced, to refine the diffusion process.

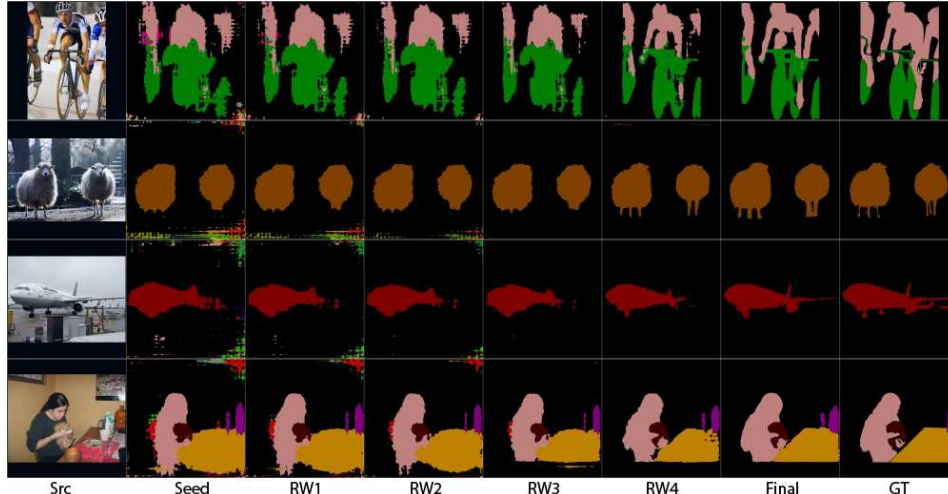

| Src | Seed | RW1 | RW2 | RW3 | RW4 | Final | GT |

Figure 4: Visualization of our seed and outputs after each random walk $\oplus$.

**Similarity Branch**   Our cascaded random walks are carried out on a sequence of transition matrices $P_t$ which measure similarities on the different level of semantics. To visualize these hierarchical semantic similarities, in Fig. 3, for selected node $i$, we reshape the corresponding rows in all the transition matrices $P_{t_{i,:}}$ to $[h', w']$ and show them by color coding. $P_{t_{i,:}}$ represents the possibilities that other nodes random walk to node $i$ based on $t$-th transition matrix with $\sum_j P_{t_{i,j}} = 1$. As shown in Fig. 3, from $P_1$ to $P_5$, similarities are measured from low-level feature such as color, texture to high-level feature such as object. Particularly, $P_t$ is able to identify fine-grained similarities among pixels belonging to coarsely labeled objects, such as figures for nodes $1, 6$ where table mat and painting are highlighted when they are labeled as table and background respectively. These results also meet our assumption that similarity branch estimates the possibility of any two nodes that belong to the same class without knowing which class they are. Finally, figures for nodes $3, 4, 6, 7$ also prove the ability of capturing long-range dependencies in our model.

**Diffusion**   We show outputs of random walks in Fig. 4. $\text{RW}_t$ represents the output after $t$-th random walk $\oplus_t$. Obviously, the outputs are gradually refined after each random walk. We also report learned $\mu_t$ and $\beta_t$ for each random walk in Table. 3. The increasing of parameter value means the output is more depended on information transited from other nodes rather than initial seed and previous random walk result as data flows through our model. To validate the effectiveness of the transition matrices built on all the ResNet blocks, we also test DifNet-50 without 2-th and 4-th random walks, the performance will have 1 percent drop on Pascal VOC validation dataset.

|          | $\mu_t$  | $\beta_t$ |
|----------|----------|-----------|
| $\oplus_1$ | 0.4159 | 0.4159 |
| $\oplus_2$ | 0.4193 | 0.4825 |
| $\oplus_3$ | 0.4077 | 0.5104 |
| $\oplus_4$ | 0.6520 | 0.6570 |
| $\oplus_5$ | 0.8956 | 0.8451 |

Table 3: Learned $\mu_t$ and $\beta_t$ for each $\oplus_t$.

| model | | time |
|-------|--|------|
| DifNet-50      Seed | | 0.018±0.003s |
|             Similarity | | 0.015±0.003s |
|             Diffusion | | 0.006±0.001s |
| Sim-Deeplab-101 | | 0.036±0.003s |

Table 4: Time consumption comparison.

## 5.4 Efficiency Study

In Table. 4, we report the time consumption for doing inference with inputs of size $[3, 505, 505]$ on one GTX 1080 GPU. For inference, total time consumption of our DifNet-50 is equivalent to Sim-Deeplab-101. However, in this case, the data can flow through two branches of our model parallelly, so the computation of our model can be further accelerated by model parallel to two times faster. The diffusion process only involves matrix multiplication (five random walks) and can be implemented efficiently with little extra computation.

On the contrary, the backpropagation of our model will require much more calculations compared with the vanilla model. Since the outputs of two branches determine the final results together from amounts of information propagation paths, the parameters of two branches will be heavily mutually influenced when doing optimization. The time consumption of backpropagation in our DifNet-50 model is about 1.3 times than Sim-Deeplab-101. However, in view of benefits from model parallel during inference, extra time spent on training is considered acceptable.

# 6    Conclusion

We present DifNet for semantic segmentation task, our model applies the cascaded random walks to approximate a complex diffusion process. With these cascaded random walks, more details can be complemented according to the hierarchical semantic similarities and meanwhile long-range dependencies are captured. Our model achieves promising performance compared with various baseline models, the effectiveness of each component in our model is also verified through comprehensive mechanism studies.

# Acknowledgment

This work was supported by the grants of National Natural Science Foundation of China (61702301), China Postdoctoral Science Foundation funded project (2017M612272), Fundamental Research Funds of Shandong University, National Natural Science Foundation of China (61332015) and National Basic Research grant (973) (2015CB352501).

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
