[Reviews · NeurIPS 2018]

Reviewer 1



This paper proposes a method for semantic image segmentation in which a single network integrates initial per-pixel class predictions from a DeepLab V2[7]-based network with pairwise affinities derived from learned per-pixel embeddings. The final predictions of the network are obtained by updating the initial predictions using a series of matrix multiplies by transition matrices computed efficiently from the pairwise affinities. The main contribution of the work seems to be the use of multiple embeddings, rather than a single one, and the use of a more computationally-efficient inference procedure than some, but not all, prior works. The authors evaluate the proposed method on the Pascal VOC and Pascal Context datasets, and compare to the Deeplab V2 [7] approach on which the proposed method is based. Strengths: - The entire network is trained end-to-end, directly with a cross-entropy loss on the final predictions; in contrast, prior work has relied on more complicated, ad-hoc training strategies, although in many cases still end-to-end. - DeepLab V2 is a reasonable baseline for comparison. - The Pascal VOC 2012 and Pascal Context datasets are reasonable choices for evaluation. - Proposed method is much more computationally efficient than DeepLab V2 with the CRF step included, and is also more computationally efficient than a similar prior work [3]. Weaknesses: - While the proposed method outperforms certain restricted variants of DeepLab V2 [7], it performs significantly worse than the full DeepLab V2 on both of the two datasets used for comparisons. - The experimental results show that the proposed method outperforms a restricted variant of DeepLab V2 that does not include the MSC, COCO, and CRF improvements (that are also not used with the proposed method). However, the proposed method has worse accuracy than the full DeepLab V2 approach, with a particularly large accuracy gap on the Pascal VOC 2012 dataset. The authors should give results for the proposed method in combination with the MSC, COCO, and CRF improvements. - The proposed method is very similar to two prior works cited by the authors [3, 5]. The paper should more clearly compare the proposed method to these prior works, and include them for comparison in the tables of experimental results. - The proposed method seems to have lower accuracy than the two very similar prior works [3] and [5], and also seems to offer no advantage in terms of computational efficiency over [5]. - The paper would benefit from proofreading of the grammar. Response to author rebuttal: I thank the authors for clarifying the computational complexity of the diffusion step, specifically the fact that it is done with the features downsampled to 64x64; this addresses my confusion regarding the reported computation times. I appreciate the difficulty of reproducing and comparing against [3] and [5] without source code, and that the that they may also be more difficult to train than the proposed method. While a more robust method is a valuable contribution even if it does not provider better accuracy or speed, I think a more thorough investigation than is described in the paper would be needed to show that. Regarding the lack of evaluation with the DeepLab MSC, COCO and CRF steps, while the authors make the valid point that the MSC step and especially the CRF step increase the computational complexity, and that the COCO pretraining adds training data and therefore it may not be fair to compare COCO-pretrained models to those without it, evaluations including these steps are nonetheless important for properly comparing this method to other methods, given the existing similar work.

Reviewer 2



Summary: A seed branch is used to generate semantic seeds and a similarity branch which computes pixel similarities. A cascade random walk neural network is designed to diffuse information regarding the constructed graph considering the affinity matrix. Originality: The author should highlight its difference with the method proposed in "Convolutional Random Walk Networks for Semantic Image Segmentation" which also use similarity method. And I think a major difference is the similarity learning and cascade random walk which I think the inherent idea is very similar. Significance: Quite interesting and graph-based method for semantic segmentation is a very interesting direction.

Reviewer 3



The paper proposes a novel two stage scheme for semantic segmentation. In the first stage, a sub-network predicts the initial rough segmentation results which are the seed of the second stage. In the second stage, another sub-network estimates the similarities between the pixels based on their multi-level features. Finally, cascaded random walk is performed to update the results. The evaluations on Pascal VOC 2012 and Pascal Context validate the effectiveness of the proposed method combined with Deeplab v2. The novelty of the proposed method is sufficient as common segmentation networks are purely feed-forward ones, e.g. PSPnet and Deeplab. The proposed network addresses the boundary localization problem and fragmented prediction simultaneously within a single framework. Compared to existing networks that incorporate crf and random walk, the proposed method seems still unique. One concern is that the proposed method should be compared to Deeplab v2 that contain crf or random walk components described in previous papers, in terms of mIoU. Comparison to only Deeplab v2 itself is not sufficient.